# Electroencephalography (EEG) based epilepsy diagnosis via multiple feature space fusion using shared hidden space-driven multi-view learning

Xiujian Hu[1], Yicheng Xie[1], Hui Zhao[1], Guanglei Sheng[1,2], Khin Wee Lai[3] and Yuanpeng Zhang[4]

[1] Department of Electronics and Information Engineering, Bozhou University, Bozhou, Anhui, China
[2] School of Computer Science and Engineering, Xi'an University of Technology, Xi'an, Shanxi, China
[3] Department of Biomedical Engineering, Universiti Malaya, Kuala Lumpur, Malaysia
[4] Department of Medical Informatics, Nantong University, Nantong, Jiangsu, China



## ABSTRACT

Epilepsy is a chronic, non-communicable disease caused by paroxysmal abnormal synchronized electrical activity of brain neurons, and is one of the most common neurological diseases worldwide. Electroencephalography (EEG) is currently a crucial tool for epilepsy diagnosis. With the development of artificial intelligence, multi-view learning-based EEG analysis has become an important method for automatic epilepsy recognition because EEG contains difficult types of features such as time-frequency features, frequency-domain features and time-domain features. However, current multi-view learning still faces some challenges, such as the difference between samples of the same class from different views is greater than the difference between samples of different classes from the same view. In view of this, in this study, we propose a shared hidden space-driven multi-view learning algorithm. The algorithm uses kernel density estimation to construct a shared hidden space and combines the shared hidden space with the original space to obtain an expanded space for multi-view learning. By constructing the expanded space and utilizing the information of both the shared hidden space and the original space for learning, the relevant information of samples within and across views can thereby be fully utilized. Experimental results on a dataset of epilepsy provided by the University of Bonn show that the proposed algorithm has promising performance, with an average classification accuracy value of 0.9787, which achieves at least 4% improvement compared to single-view methods.

## INTRODUCTION

Epilepsy is a chronic, non-infectious but genetic disease that affects all ages and is caused by paroxysmal abnormal hypersynchrony of brain neurons. It is one of the most common neurological diseases globally. Due to the diversity and complexity of the clinical manifestation of epilepsy, it is often misdiagnosed or missed. Repetitive seizures can have a

Corresponding author
Yuanpeng Zhang,
maxbirdzhang@ntu.edu.cn

persistent negative impact on the patient's mental and cognitive functions, even threatening their life. Therefore, the study of epilepsy diagnosis and treatment has important clinical significance. The brain electroencephalogram (EEG) is a microvolt-level electrical signal generated by synchronized neurons in the brain when electrodes are placed on the scalp at specific locations. As the most commonly used and cheapest non-invasive brain wave detection method, EEG has a history of over 70 years of research and is the most effective method for diagnosing epilepsy-related diseases, such as identifying seizures, predicting their occurrence, and localizing the affected areas. With the development of artificial intelligence, machine learning models are extensively used in automatic epilepsy recognition. Feature representation is a crucial step in machine learning. Research has indicated that EEG signals can be represented by both linear and non-linear features. Time-domain features are the fundamental features in EEG signal processing, primarily extracted by directly observing and calculating relevant characteristics from the raw signal. Their advantages lie in their simplicity of computation and ease of interpretation. However, the non-stationarity of EEG signals, individual differences, and external interferences can easily affect time-domain features. Frequency-domain features are based on the significant changes in energy in EEG during epileptic seizures, assuming that the background EEG is approximately stationary. Most frequency-domain features are derived from the study of signal power spectra, and various parameter estimation methods can be used for extracting spectral features. The accuracy of these parameters also affects the quality of frequency-domain features. If we consider the amount of information contained in the features, neither pure time-domain features nor frequency-domain features can comprehensively characterize an EEG signal. Additionally, EEG analysis based on the assumption of stationarity is not rigorous. Therefore, researchers have turned their attention to time-frequency analysis methods, such as time-frequency transformations, to re-represent non-stationary EEG signals and extract corresponding features. In addition to the aforementioned linear features, many studies also consider the brain as a nonlinear system and extract corresponding nonlinear features from descriptions of complexity, persistence, synchrony, and other changes in the system. These features are not affected by the non-stationarity of EEG signals and offer more flexibility in dealing with issues such as multi-channel correlation and channel loss. Based on the aforementioned linear or nonlinear feature representations, numerous scholars have constructed machine learning models for the automatic diagnosis of epilepsy. For example, the study conducted by *Li, Chen & Zhang (2016)* employed a dual-tree complex discrete wavelet transform to extract nonlinear features from individual components. The researchers utilized an ANOVA analysis to select relevant classification features, including the Hurst parameter and fuzzy entropy. For the classification task, a support vector machine (SVM) was employed. *Reddy & Rao (2017)* computed the central correlated entropy of wavelet components obtained from tunable Q-factor wavelet transform, and utilized models such as RF, LR, and multi-layer perceptron for epileptic signal recognition. *Jaiswal & Banka (2017)* proposed a feature extraction method called local gradient pattern transformation and applied classification methods such as k-nearest neighbors, SVM, and decision trees for epilepsy detection.

The aforementioned machine learning-based epilepsy diagnostic models utilize single EEG feature representation for epilepsy diagnosis, which have low model complexity and high interpretability. However, these models rely on expert knowledge, and deep features are not easily observed and extracted. As a result, the accuracy is limited. Multi-view learning (*Zhao et al., 2017*; *Jiang et al., 2020*; *Zhang, Chung & Wang, 2018*; *Yan et al., 2021*) improves the classification accuracy of models by utilizing the differences and similarities between multiple different views based on the principles of view consistency and complementarity. For example, *Tian et al. (2019)* utilized a convolutional neural network (CNN) model to extract deep features from EEG signals in the time domain, frequency domain, and time-frequency domain. These features were constructed as three views, and multi-view learning was conducted using a multi-view Takagi-Sugeno-Kang (TSK) fuzzy system, which improved the classification and detection performance compared to a single view. *Yuan et al. (2018)* implemented a multi-view epilepsy automatic diagnosis by utilizing channel characteristics and intra-channel time-frequency features of multi-channel EEG signals extracted using autoencoder (AE) through channel perception technology. *Liu & Li (2019)* utilized a user-sensitive model for channel selection and extracted time-frequency features from each sub-band of the selected channels, forming multi-view features. They extracted numerical and morphological features using a common spatial projection matrix and utilized a maximum average difference autoencoder to extract inter-channel time-frequency domain features, enabling automatic diagnosis of epilepsy with multiple views. These effective models based on collaborative regularization can construct a common feature space for multi-view learning. However, these models also have certain limitations. While these methods construct the density distributions of each view solely based on the corresponding observed data, they overlook the correlated information among all views. Additionally, they separate the original sample space from the common space obtained through mapping. This approach solely utilizes the common space for learning, neglecting the discriminative information present in the original space.

To overcome such shortcomings, in this study, a shared hidden feature space method is constructed by using kernel density estimation, and it is extended to an expanded space by combining it with the original space. Then, SVM is introduced and a multi-view SVM based on the shared hidden space is proposed to take a careful consideration of the differences and relationships between samples from different views. Through experimental verification on different multi-view data sets, the effectiveness of this method in addressing the challenges mentioned above has also been confirmed. The contributions of this study are mainly reflected in the following aspects:

(1) The kernel density estimation (KDE) technique is used to construct a new shared hidden space, and it is combined with the original space to construct an expanded space for multi-view learning, thus being able to effectively address the special issue mentioned above on multi-view learning.

(2) By constructing the expanded space and utilizing the information of both the shared hidden space and the original space for learning, thereby fully utilizing the relevant information of samples within and across views, we can effectively solve the problem that

the difference between samples of the same class from different views is greater than the difference between samples of different classes from the same view.

(3) During the optimization phase, the proposed model is transformed into a classical Quadratic Programming (QP) problem, allowing for the utilization of pre-existing optimization methods that offer both high effectiveness and theoretical guarantees. This transformation enables the application of readily available optimization techniques, which have proven to be highly efficient in solving QP problems.

The following sections are organized as follows. In 'Data', we introduce the EEG data used in this study and the corresponding multiple feature space representation. In 'Methodology', we present the proposed model. In 'Experimental studies', experimental results are reported and in the last section, the whole study is summarized.

## DATA

The EEG data of epileptic patients used in this study was authorized and provided by the University of Bonn in Germany (*Andrzejak et al., 2001*), as shown in Table 1. The dataset included volunteers who could be divided into five groups, namely A, B, C, D, and E. Each group contained 100 single-channel EEG segments lasting 23.6 s, with a sampling rate of 173.6 Hz. The EEG signals of groups A and B were collected from healthy volunteers in a relaxed and conscious state, while the eyes of the volunteers were open during the data collection of group A and closed during the data collection of group B. The remaining three groups' signals were collected from epileptic volunteers, with group C's signals collected from the hippocampi of the two brain hemispheres, and group D's signals collected from the epileptic foci. The signals of groups C and D were measured during periods without epileptic seizures, while group E collected signals during epileptic seizures. Figure 1 provides an example of EEG signals from five groups.

### Frequency-domain representation extraction

Frequency-domain feature representation originates from the significant changes in energy in EEG during epileptic seizures. To extract frequency-domain representation from EEG signals, the Daubechies4 wavelet coefficients are utilized to decompose the original signals into a series of binary wavelets. The frequency band of each Daubechies4 wavelet coefficient is provided in Table 2. By applying these settings, the EEG signals are divided into six distinct frequency bands. An illustrative example of the decomposed signals from group E is depicted in Fig. 2.

### Time-domain feature extraction

Time-domain features are the fundamental features in EEG signal processing, primarily extracted by directly observing and calculating relevant characteristics from the raw signal. Their advantages lie in their simplicity of computation and ease of interpretation for researchers. In this study, we employ kernel principal component analysis (KPCA) (*Li et al., 2022b*) on the raw EEG signals to enable complex nonlinear mapping. Previous research has shown that KPCA features offer discriminative patterns suitable for pattern

**Table 1 Basic collection information of epilepsy EEG signals.**

| Group | #Volunteers | Collection information |
|---|---|---|
| A | 100 | This group was collected from a group of healthy volunteers who were instructed to keep their eyes open during the recording process. These volunteers did not have any known neurological or psychiatric disorders and were not experiencing any abnormal symptoms at the time of data collection. |
| B | 100 | This group was collected from a group of healthy volunteers under conditions where they kept their eyes closed. |
| C | 100 | This group was collected from the hippocampal formation of the contralateral hemisphere of the brain during seizure-free intervals. These samples were obtained when the patient was not experiencing any epileptic seizures. |
| D | 100 | This group was collected from the epileptogenic zone during periods of seizure freedom. This implies that the recordings were obtained when the patient was not experiencing seizures. |
| E | 100 | The group was collected during seizure activity phase offering a unique opportunity to study the dynamics and temporal dynamics of epileptic seizures, paving the way for the development of more accurate and reliable seizure detection and prediction algorithms. |

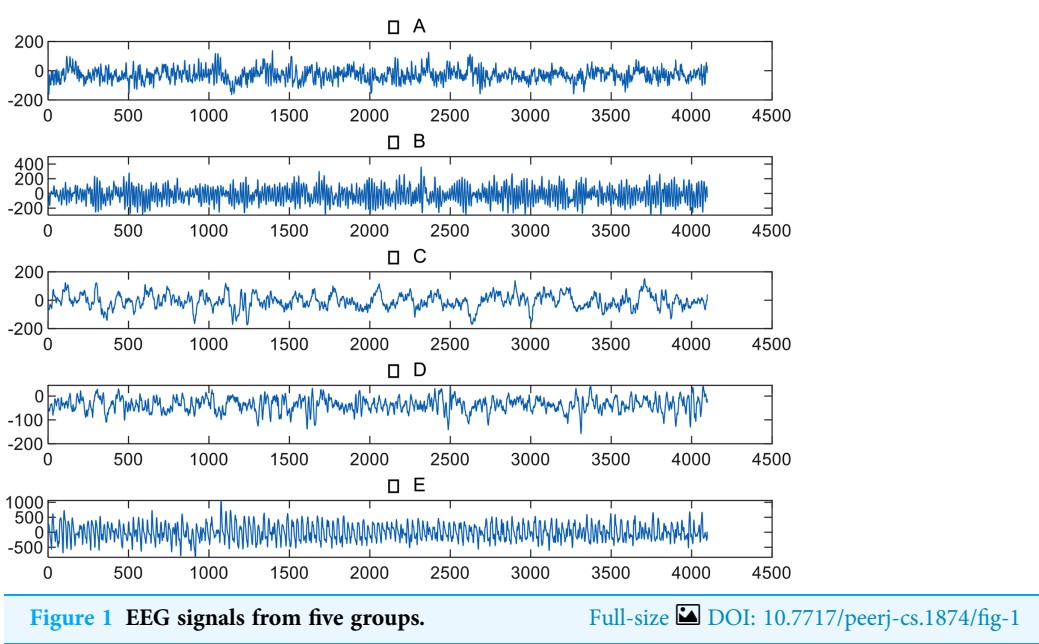

**Figure 1 EEG signals from five groups.**

**Table 2 Frequency band of each Daubechies4 wavelet coefficient.**

| Coefficient | Frequency band |
|---|---|
| Daubechies4 (4, 0) | 0–2 Hz |
| Daubechies4 (4, 5) | 2–4 Hz |
| Daubechies4 (4, 4) | 4–8 Hz |
| Daubechies4 (4, 3) | 8–15 Hz |
| Daubechies4 (4, 2) | 16–30 Hz |
| Daubechies4 (4, 1) | 31–60 Hz |

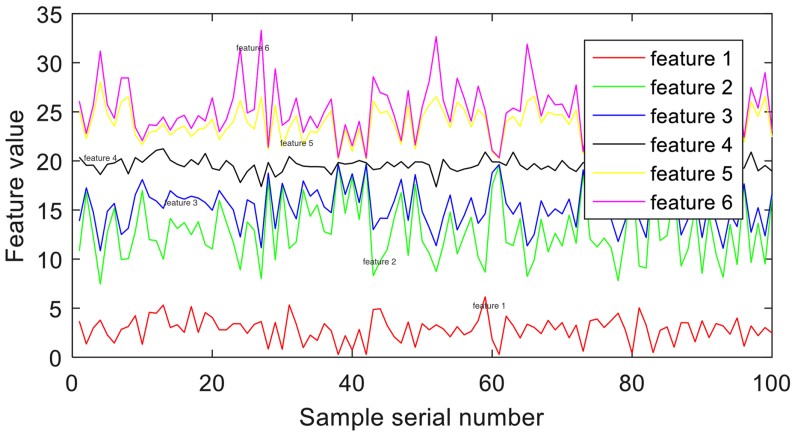

**Figure 2  Example of frequency-domain representation.**

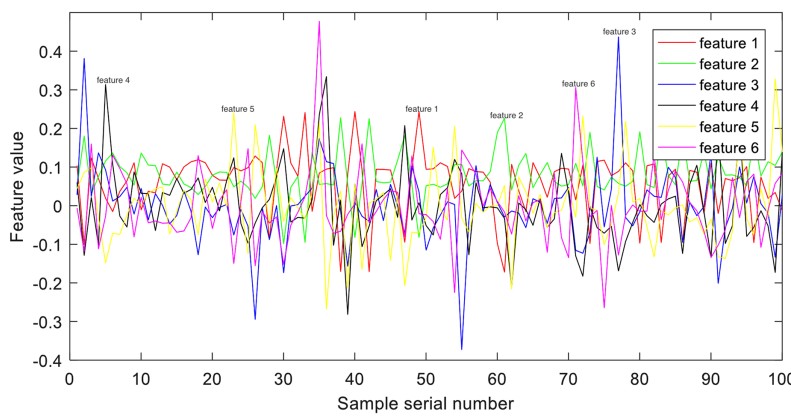

**Figure 3  Example of time-domain representation.**    

recognition. An illustration depicting an example of KPCA features from group E can be observed in Fig. 3.

## Time-frequency representation extraction

Pure time-domain or frequency-domain feature representations alone cannot comprehensively characterize an EEG signal, and EEG analysis based on the assumption of stationarity is not rigorous. Therefore, researchers have turned their attention to time-frequency analysis methods, such as time-frequency transformations, to re-represent non-stationary EEG signals and extract corresponding features. To capture time-frequency representation, researchers often employ the short-time Fourier transform (STFT) (*Li et al., 2022a*). STFT allows for the analysis of how the frequency content of a signal changes over time. It can be formulated as follows:

$$F_{time-fre}(time, fre) = \int_{-inf}^{+inf} x(time)g(time - u)e^{-j2\pi*fre*time}d(time). \tag{1}$$

In the context of EEG signal analysis, Eq. (1) represents the transformation of continuous EEG signals, denoted as $x(time)$, into the time-frequency plane using the

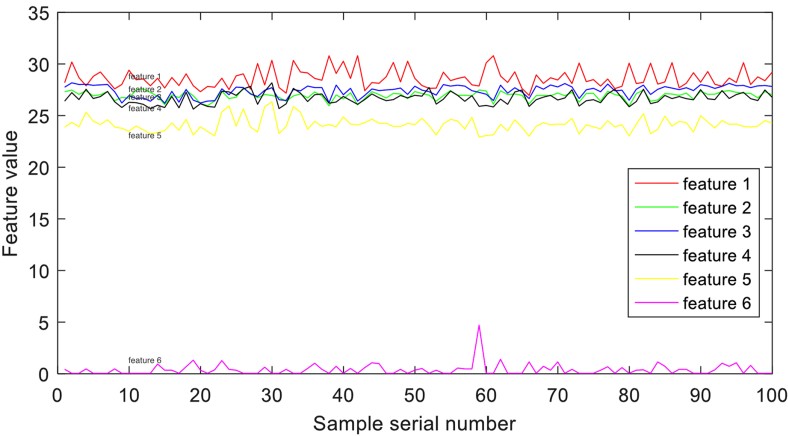

**Figure 4** **Example of time-frequency representation.**

function $g(time - u)$ and a limited width window centered around $u$. This transformation, referred to as $F_{time-fre}(time, fre)$, provides a means to examine the time-varying nature of the EEG signals, revealing local spectrum discrepancies at different time points. To achieve this, the EEG signals undergo partitioning into several segments of local stationary signals using STFT. Through this process, the time-varying characteristics of the EEG signals are captured, highlighting variations in the spectrum. The extraction of six energy bands as features is accomplished using Eq. (1), which takes into account the observed discrepancies. A visualization of these six energy bands, exemplified by group E, is illustrated in Fig. 4.

## METHODOLOGY

In this section, we will design a shared hidden space-driven multi-view learning method to fuse time-frequency representation, frequency-domain representation and time-domain representation.

### Construction of shared hidden feature space

Suppose that $\Omega \in \boldsymbol{R}^{r \times d}$ is an orthogonal matrix subject to $\Omega \Omega^T = \boldsymbol{I} \in \boldsymbol{R}^{r \times r}$, $f^A = \{\mathbf{x}_i^A, y_i | \mathbf{x}_i^A \in R^d, i = 1, 2, \ldots, N\}$ represents one kind of feature space, *e.g.*, time-domain feature space, and $f^B = \{\mathbf{x}_i^B, y_i | \mathbf{x}_i^B \in R^d, i = 1, 2, \ldots, N\}$ represents another kind of feature space, then the hidden feature space of $f^A$ and $f^B$ can be generated by $\Omega x_i^A \in \boldsymbol{R}^r$ **and** $\Omega x_i^B \in \boldsymbol{R}^r$, respectively, where $r$ represents the number of hidden features. To obtain a consistent hidden feature space between $\Omega x_i^A$ and $\Omega x_i^B$, it is expected that the difference between them should be minimized as much as possible. Kernel density estimation (KDE), which is one of the non-parametric estimation methods in probability theory, is usually used to estimate the unknown probability density function (*Wang, Wang & Chung, 2013*). For a training set $X = \{\mathbf{x}_i, y_i | \mathbf{x}_i \in R^d, i = 1, 2, \ldots, N\}$, its corresponding kernel density estimation function can be expressed as

$$P(\mathbf{x}) = \frac{1}{N} \sum_{i=1}^{N} \delta^2 K\left(\frac{\mathbf{x} - \mathbf{x}_i}{\delta}\right), \quad (2)$$

where $\delta$ is the kernel width, $K(\cdot)$ is the kernel function. If the Gaussian kernel function is adopted, then Eq. (2) can be updated as $P(\mathbf{x}) = \frac{1}{N}\sum_{i=1}^{N}\frac{1}{\delta\sqrt{2\pi}}\exp\left(-\frac{1}{2}\left(\frac{\mathbf{x}-\mathbf{x}_i}{\delta}\right)^2\right)$.

Therefore, the kernel density estimation of $\Omega x_i^A$ and $\Omega x_i^B$ can be expressed as follows when using the Gaussian kernel function, respectively,

$$-\frac{\|\Omega x - \Omega x_i^A\|^2}{2\delta^2} \quad P_A(\tilde{\mathbf{x}}) = P_A(\Omega x) = \frac{1}{N\cdot\delta\sqrt{2\pi}}\sum_{i=1}^{N}\mathrm{e}, \tag{3}$$

$$-\frac{\|\Omega x - \Omega x_i^B\|^2}{2\delta^2} \quad P_B(\tilde{\mathbf{x}}) = P_B(\Omega x) = \frac{1}{N\cdot\delta\sqrt{2\pi}}\sum_{i=1}^{N}\mathrm{e}. \tag{4}$$

In this study, the difference between $P_A(\tilde{\mathbf{x}})$ and $P_B(\tilde{\mathbf{x}})$ is measured by the mean square error, that is

$$J = \int(P_A(\tilde{\mathbf{x}}) - P_B(\tilde{\mathbf{x}}))^2 \mathrm{d}\mathbf{x}. \tag{5}$$

By minimizing $J$, the two-view data $\mathbf{x}_i^A$ and $\mathbf{x}_i^B$ can be made to have the maximum commonality in the shared hidden space, and thus the challenge of excessive variability between samples from different views can be addressed. In order to solve Eq. (6), we suppose that $G(\Omega x, \Omega x_i, \delta^2) = \frac{1}{\delta\sqrt{2\pi}}\mathrm{e}^{-\frac{\Omega x - \Omega x_i^2}{2\delta^2}}$, then $P_A(\tilde{\mathbf{x}})$ and $P_B(\tilde{\mathbf{x}})$ can be updated as $P_A(\tilde{\mathbf{x}}) = \frac{1}{N}\sum_{i=1}^{N}G(\Omega x \Omega x_i^A, \delta^2)$ and $P_B(\tilde{\mathbf{x}}) = \frac{1}{N}\sum_{i=1}^{N}G(\Omega x \Omega x_i^B, \delta^2)$. Therefore, Eq. (5) can be computed by $J = \int P_A(\tilde{\mathbf{x}})\mathrm{d}x - 2\int P_A(\tilde{\mathbf{x}})P_B(\tilde{\mathbf{x}})\mathrm{d}x + \int P_B(\tilde{\mathbf{x}})\mathrm{d}x$. According to *Wang, Wang & Chung (2013)*, *Hansen, Jaumard & Xiong (1994)*, we have $\int G(\mathbf{x}, \mathbf{x}_i, \delta_1^2)G(\mathbf{x}, \mathbf{x}_j, \delta_2^2)dx = G(\mathbf{x}_i, \mathbf{x}_j, \delta_1^2 + \delta_2^2)$, Therefore, we have the following equations,

$$\int P_A^2(\tilde{\mathbf{x}})dx = \frac{1}{N^2}\sum_{i=1}^{N}\sum_{j=1}^{N}G(\tilde{\mathbf{x}}_i^A, \tilde{\mathbf{x}}_j^A, 2\delta^2) = \frac{1}{N}\sum_{i=1}^{N}\left[\frac{1}{N}\sum_{j=1}^{N}G(\tilde{\mathbf{x}}_i^A, \tilde{\mathbf{x}}_j^A, 2\delta^2)\right] \tag{6}$$

$$\int P_B^2(\tilde{\mathbf{x}})dx = \frac{1}{N^2}\sum_{i=1}^{N}\sum_{j=1}^{N}G(\tilde{\mathbf{x}}_i^B, \tilde{\mathbf{x}}_j^B, 2\delta^2) = \frac{1}{N}\sum_{i=1}^{N}\left[\frac{1}{N}\sum_{j=1}^{N}G(\tilde{\mathbf{x}}_i^B, \tilde{\mathbf{x}}_j^B, 2\delta^2)\right] \tag{7}$$

$$\int P_A(\tilde{\mathbf{x}})P_B(\tilde{\mathbf{x}})dx = \frac{1}{N^2}\sum_{i=1}^{N}\sum_{j=1}^{N}G(\tilde{\mathbf{x}}_i^A, \tilde{\mathbf{x}}_j^B, 2\delta^2) \tag{8}$$

where $\frac{1}{N}\sum_{j=1}^{N}G\left(\tilde{\mathbf{x}}_i^A, \tilde{\mathbf{x}}_j^A, 2\sigma^2\right)$ can be taken as another estimation of $P_A(\tilde{\mathbf{x}}_i^A)$. Therefore, $\int P_A^2(\tilde{\mathbf{x}})\mathrm{d}x$ can be estimated by $\frac{1}{N}\sum_{j=1}^{N}P_A(\tilde{\mathbf{x}}_i^A)$, and further $\frac{1}{N}$. Similarly, $\int P_B^2(\tilde{\mathbf{x}})\mathrm{d}x$ can be estimated by $\frac{1}{N}$. Thus, we finally have $J \approx \frac{1}{N} + \frac{1}{N} - \frac{2}{N^2}G\left(\tilde{\mathbf{x}}_i^A, \tilde{\mathbf{x}}_j^B, 2\delta^2\right)$. Therefore, we have the following objective,

$$\arg\min_{\Omega} J \approx \arg\min_{\Omega} \sum_{i=1}^{N}\sum_{j=1}^{N}G\left(\tilde{\mathbf{x}}_i^A, \tilde{\mathbf{x}}_j^B, 2\delta^2\right) \tag{9}$$

$$s.t.\,\Omega\Omega^T = I_{r\times r}$$

However, it is difficult to solve Eq. (9) directly. Thus, Taylor expansion can be used for getting an approximate solution. Hence, we have

$$G\left(\tilde{\mathbf{x}}_i^A, \tilde{\mathbf{x}}_j^B, 2\delta^2\right) = \frac{1}{\sqrt{2\pi}\delta} e^{-\frac{\Omega x_i^A - \Omega x_j^{B^2}}{4\sigma^2}} \approx \frac{1}{\sqrt{2\pi}\delta}\left(1 - \left(\Omega x_i^A - \Omega x_j^B\right)^2\right) \tag{10}$$

Therefore, Eq. (9) can be further updated as

$$\arg\min_{\Omega} \sum_{i=1}^{N}\sum_{j=1}^{N}\left(\Omega x_i^A - \Omega x_j^B\right)^2, s.t.\Omega\Omega^T = \mathbf{I}_{r\times r} \tag{11}$$

in Eq. (11), implicit feature transformation matrix $\Omega$ still cannot be solved directly, but can be solved by gradient descent method. Thus, Eq. (11) can be updated as

$$J = \underset{\Omega}{\mathrm{argmin}} \sum_{i=1}^{N}\sum_{j=1}^{N}\left(\left(\mathbf{x}_i^A\right)^T\Omega^T\Omega x_i^A + \left(\mathbf{x}_j^B\right)^T\Omega^T\Omega x_j^B - 2\left(\mathbf{x}_i^A\right)^T\Omega^T\Omega x_j^B\right) \tag{12}$$

$$s.t. \quad \Omega\Omega^T = \mathbf{I}_{r\times r}$$

The partial derivative of $J$ w.r.t. $\Omega$ is

$$\frac{\partial J}{\partial \Omega} = \sum_{i=1}^{N}\sum_{j=1}^{N}\left(2\Omega x_i^A\left(\mathbf{x}_i^A\right)^{\mathrm{T}} + 2\Omega x_j^B\left(\mathbf{x}_j^B\right)^{\mathrm{T}} - 2\Omega\left(\mathbf{x}_i^A\left(\mathbf{x}_i^A\right)^{\mathrm{T}} + \mathbf{x}_j^B\left(\mathbf{x}_j^B\right)^{\mathrm{T}}\right)\right) \tag{13}$$

Then the transformation matrix $\Omega$ can be solved by gradient descent method, that is,

$$\Omega \leftarrow \Omega - \eta\frac{\partial J}{\partial \Omega}\left(\mathbf{I}_{r\times r} - \Omega\Omega^T\right) = \Omega - \eta\nabla\Omega \tag{14}$$

where $\eta$ is the step size that can be solved by

$$\eta = \sum_{i=1}^{N}\sum_{j=1}^{N}\left(\left(\mathbf{x}_i^A\right)^{\mathrm{T}}\left(\Omega^T\nabla\Omega + \nabla\Omega^T\Omega\right)\mathbf{x}_i^A + \left(\mathbf{x}_j^B\right)^{\mathrm{T}}\left(\Omega^T\nabla\Omega + \nabla\Omega^T\Omega\right)\mathbf{x}_j^B\right.$$

$$\left. - \frac{2\left(\mathbf{x}_i^A\right)^{\mathrm{T}}\left(\Omega^T\nabla\Omega + \nabla\Omega^T\Omega\right)\mathbf{x}_j^B}{\sum_{i=1}^{N}\sum_{j=1}^{N}\left(2(\mathbf{x}_i^A)^{\mathrm{T}}\nabla\Omega^T\nabla\Omega x_i^A\right)} + \left(\mathbf{x}_j^B\right)^{\mathrm{T}}\nabla\Omega^T\nabla\Omega x_j^B - 4\left(\mathbf{x}_i^A\right)^{\mathrm{T}}\nabla\Omega^T\nabla\Omega x_j^B\right) \tag{15}$$

According to the above analysis and derivation, the algorithm for solving implicit feature transformation matrix $\Omega$ is described as follows.

## Multi-view learning based on shared hidden feature space

After determining the shared hidden space between two views, the extended space can be generated by combining the original space and the shared hidden space. Then, a multi-view classifier based on SVM is designed for multi-view data classification in the extended

space. In existing multi-view learning mechanisms, it is generally assumed that each view can provide a classifier containing specific information, and classifiers constructed from different view tend to be consistent. Additionally, since views can provide specific information to each other, the proposed model establishes the objective function by considering the mutual information between two views. In summary, the proposed model, based on SVM, restructures the slack variables on each view, and then narrows the gap between the two views by using the corresponding regularization term. The objective function of multi-view learning based on shared hidden feature space can be formulated as

$$arg \min_{\mathbf{w}_A,\mathbf{w}_B,\mathbf{v}_A,\mathbf{v}_B,b_A,b_B} \frac{1}{2} \parallel \mathbf{w}_A \parallel^2 + \frac{1}{2} \parallel \mathbf{w}_B \parallel^2 + \frac{1}{2} \parallel \mathbf{v}_A \parallel^2 + \frac{1}{2} \parallel \mathbf{v}_B \parallel^2 + C^A \sum_{i=1}^{N} \xi_i^A$$

$$+ C^B \sum_{i=1}^{N} \xi_i^B + \lambda \parallel \mathbf{v}_A - \mathbf{v}_B \parallel^2 \tag{16}$$

$$s.t. y_i(w_A^T \phi(\mathbf{x}_i^A) + v_A^T \phi(\Omega x_i^A) + b_A) \geq 1 - \xi_i^A$$
$$y_i(w_B^T \phi(\mathbf{x}_i^B) + v_B^T \phi(\Omega x_i^B) + b_B) \geq 1 - \xi_i^B$$
$$\xi_i^A, \xi_i^B \geq 0, i = 1, 2, \ldots, N$$

where $\lambda$, $C^A$ and $C^B$ are the regularization parameters. Observe that Eq. (16) consists of three parts: the first four terms reflect the outcome risk in the original feature space and the shared hidden space respectively; the second two terms represent the empirical risk; and the third term reflects the difference between the two views in the shared hidden space. The objective function in Eq. (16) strengthens the constraints based on the traditional SVM through the implicit mapping, so that the probability distributions of data from different views in the shared hidden space are as consistent as possible, which can well solve the problem described at the beginning of this study. In order to solve Eq. (16) efficiently, the relevant Lagrangian multipliers are introduced according to the Lagrangian optimization theory, hence Eq. (16) can be converted into the corresponding dual form as follows. The Lagrangian function corresponding to Eq. (16) is

$$L = \frac{1}{2} \parallel \mathbf{w}_A \parallel^2 + \frac{1}{2} \parallel \mathbf{w}_B \parallel^2 + \frac{1}{2} \parallel \mathbf{v}_A \parallel^2 + \frac{1}{2} \parallel \mathbf{v}_B \parallel^2 + C^A \sum_{i=1}^{N} \xi_i^A$$

$$+ C^B \sum_{i=1}^{N} \xi_i^B + \lambda \parallel \mathbf{v}_A - \mathbf{v}_B \parallel^2$$

$$+ \sum_{i=1}^{N} \alpha_i^A (1 - \xi_i^A - y_i(w_A^T \phi(\mathbf{x}_i^A) + v_A^T \phi(\Omega x_i^A) + b_A)) \tag{17}$$

$$+ \sum_{i=1}^{N} \alpha_i^B (1 - \xi_i^B - y_i(\mathbf{w}_B^T \phi(\mathbf{x}_i^B) + v_B^T \phi(\Omega x_i^B)$$

$$+ b_B)) - \sum_{i=1}^{N} \mu_i^A \xi_i^A - \sum_{i=1}^{N} \mu_i^B \xi_i^B$$

---

**Algorithm 1 Shared hidden feature space generation.**

Input: $\mathbf{x}_i^A$, $\mathbf{x}_i B$, and $\mathbf{y} = [y_i]_{i=1,2,\ldots,N}$

**Output:** $\Omega$

**Procedures:**

**1.** Initialize $\Omega_0 \in \mathbf{R}^{r \times d}$, $t = 0$, $iter_{max}$, $\delta = 1e - 6$.

2. Repeat:

3. $t = t + 1$.

4. Compute $\dfrac{\partial J}{\partial \Omega}$ and $\eta$ by Eqs. (13) and (15).

5. Update $\Omega(t)$ by Eq. (14).

**6.** Until $\Omega(t) - \Omega(t-1) \le \delta$ or $t > iter_{max}$

---

where $\alpha_i^A \ge 0$, $\alpha_i^B \ge 0$, $\mu_i^A \ge 0$, and $\mu_i^B \ge 0$ are Lagrangian multipliers. By setting the partial derivatives of Lagrangian function $L$ with respect to $\mathbf{w}_A$, $\mathbf{w}_B$, $\mathbf{v}_A$, $\mathbf{v}_B$, $b_A$, $b_B$, $\xi_i^A$, and $\xi_i^B$ to 0, we have

$$\mathbf{w}_A = \sum_{i=1}^{N} \alpha_i^A y_i \phi\left(\mathbf{x}_i^A\right), \; \mathbf{w}_B = \sum_{i=1}^{N} \alpha_i^B y_i \phi\left(\mathbf{x}_i^B\right), \tag{18}$$

$$\mathbf{v}_A = \frac{1 + 2\lambda}{1 + 4\lambda} \sum_{i=1}^{N} \alpha_i^A y_i \phi\left(\mathbf{x}_i^A\right) + \frac{2\lambda}{1 + 4\lambda} \sum_{i=1}^{N} \alpha_i^B y_i \phi\left(\mathbf{x}_i^B\right), \tag{19}$$

$$\mathbf{v}_B = \frac{1 + 2\lambda}{1 + 4\lambda} \sum_{i=1}^{N} \alpha_i^B y_i \phi\left(\mathbf{x}_i^B\right) + \frac{2\lambda}{1 + 4\lambda} \sum_{i=1}^{N} \alpha_i^A y_i \phi\left(\mathbf{x}_i^A\right), \tag{20}$$

$$\sum_{i=1}^{N} \alpha_i^A y_i = 0, \sum_{i=1}^{N} \alpha_i^B y_i = 0, \tag{21}$$

$$C_A = \alpha_i^A + u_i^A, \; C_B = \alpha_i^B + u_i^B \tag{22}$$

By submitting Eqs. (18–22) to Eq. (16), we have the dual problem of Eq. (24), which can be defined as

$$\arg \max_{\tilde{\alpha}} -\frac{1}{2}\tilde{\alpha}^T\tilde{\alpha} + \tilde{\alpha}^T 1 . s.t. \tilde{\alpha}^T \boldsymbol{f} = 0, \boldsymbol{f} = \left[\boldsymbol{y}^T, \boldsymbol{y}^T\right]^T \tilde{\alpha}_i 0, \forall i \tag{23}$$

where

$$\tilde{\alpha} = \left[\alpha_1^A, \alpha_2^A, \ldots, \alpha_N^A, \alpha_1^B, \alpha_2^B, \ldots, \alpha_N^B\right]^T, \tag{24}$$

$$\boldsymbol{K_A} = K\left(x^A, x^A\right)\boldsymbol{yy}^T + \frac{1 + 2\lambda}{1 + 4\lambda} K\left(\Omega x^A, \Omega x^A\right)\boldsymbol{yy}^T \tag{25}$$

$$\boldsymbol{K_B} = K\left(x^B, x^B\right)\boldsymbol{yy}^T + \frac{1 + 2\lambda}{1 + 4\lambda} K\left(\Omega x^B, \Omega x^B\right)\boldsymbol{yy}^T \tag{26}$$

$$\boldsymbol{K_{AB}} = \frac{2\lambda}{1 + 4\lambda} K\left(\Omega x^A, \Omega x^B\right)\boldsymbol{yy}^T \tag{27}$$

---

**Algorithm 2** Multi-view learning based on shared hidden feature space.

Input: training samples of view-1: $\{\mathbf{x}_i^A, y_i\}$, training samples of view-2: $\{\mathbf{x}_i^B, y_i\}$, regularized parameters $C^A$, $C^B$ and $\lambda$

Output: $\mathbf{w}_A^T$, $\mathbf{w}_B^T$, $b_A$, $b_B$, $\mathbf{v}_A$ and $\mathbf{v}_B$

Procedures:

1. Use Algorithm 1 to obtain $\Omega$

2. Use $\Omega$ to obtain the shared hidden space

3. Solve the $\tilde{\alpha}_i$ according to Eq. (23)

4. Solve the $\mathbf{w}_A^T$, $\mathbf{w}_B^T$, $b_A$, $b_B$, $\mathbf{v}_A$ and $\mathbf{v}_B$ by Eqs. (18)–(22)

5. Construct the decision function based on $\mathbf{w}_A^T$, $\mathbf{w}_B^T$, $b_A$, $b_B$, $\mathbf{v}_A$ and $\mathbf{v}_B$

---

$$K = \begin{bmatrix} K_A & K_{AB} \\ K_{AB} & K_B \end{bmatrix} \tag{28}$$

$$y = [y_1, y_2, \ldots, y_N]^T \tag{29}$$

and $K$ is the kernel function. It is obvious that the optimization of Eq. (23) can be considered as a QP problem, which can be solved according to *Deng et al. (2013)*. The decision function of the proposed model in this study is defined as

$$f(x) = \frac{1}{2} \left( \mathbf{w}_A^T \phi(\mathbf{x}^A) + v_A^T \phi(\Omega \mathbf{x}^A) + b_A + \mathbf{w}_B^T \phi(\mathbf{x}^B) + v_B^T \phi(\Omega \mathbf{x}^B) + b_B \right) \tag{30}$$

The algorithm of multi-view learning based on shared hidden feature space can be obtained, as shown in Algorithm 2. From Algorithm 2, we can find that the time complexity is mainly contributed by steps 1, 3 and 4. The time complexity of Algorithm 1 is $O(Nrd + r^2)$. The time complexity of step 3 is $O((r + d)^2)$. The time complexity of step 4 is $O(N^2)$. Therefore, the time complexity of Algorithm 2 is $O(Nrd + r^2 + (r + d)^2 + N^2)$.

## EXPERIMENTAL STUDIES

### Settings

To observe the merits of the proposed model, k-nearest neighbor (KNN) (*Liu & Liu, 2016*), support vector machine (SVM) (*Liu & Liu, 2016*), SVM2K (*Farquhar et al., 2005*), multi-view L2-SVM (MV-L2-SVM) (*Huang, Chung & Wang, 2016*), and alternative multi-view MED (AMVMED) (*Chao & Sun, 2015*) are introduced for comparison studies. Accuracy is used as the evaluation indicator in this study. SVM, SVM2K, MV-L2-SVM, and 2V-SVM-SH are all trained using a Gaussian kernel for experimentation. For all methods, ten-fold cross-validation (CV) is used to determine the optimal parameters. Table 3 provides the specific parameters and ranges used for each method. All experiments are conducted on a PC with a 16-core CPU with a clock speed of 3.40 GHz and 32 GB of memory. The programming environment was Matlab R2016a.

**Table 3 Parameter settings.**

| Method | Parameter settings |
|---|---|
| KNN | $k \in \{1, 2, 3, 4, 5, 6, 7, 8, 9, 10\}$ |
| SVM | $C \in \{2e-8, 2e-7, \ldots, 2e0, 2e1, \ldots, 2e7, 2e8\}$, $\sigma \in \{2e-8, 2e-7, \ldots, 2e0, 2e1, \ldots, 2e7, 2e8\}$ |
| SVM-2K | $C^A \in \{2e-8, 2e-7, \ldots, 2e0, 2e1, \ldots, 2e7, 2e8\}$, $C^B \in \{2e-8, 2e-7, \ldots, 2e0, 2e1, \ldots, 2e7, 2e8\}$, $D \in \{2e-5, 2e-4, \ldots, 2e0, 2e1, \ldots, 2e4, 2e5\}$, $\sigma \in \{2e-8, 2e-7, \ldots, 2e0, 2e1, \ldots, 2e7, 2e8\}$ |
| MV-L2-SVM | $C^A \in \{2e-8, 2e-7, \ldots, 2e0, 2e1, \ldots, 2e7, 2e8\}$, $C^B \in \{2e-8, 2e-7, \ldots, 2e0, 2e1, \ldots, 2e7, 2e8\}$, $\sigma \in \{2e-8, 2e-7, \ldots, 2e0, 2e1, \ldots, 2e7, 2e8\}$ |
| AMVMED | $C^A \in \{2e-8, 2e-7, \ldots, 2e0, 2e1, \ldots, 2e7, 2e8\}$, $C^B \in \{2e-8, 2e-7, \ldots, 2e0, 2e1, \ldots, 2e7, 2e8\}$, $\gamma \in \{0.1, 0.2, \ldots, 0.9\}$ |
| Proposed model | $C^A \in \{2e-8, 2e-7, \ldots, 2e0, 2e1, \ldots, 2e7, 2e8\}$, $C^B \in \{2e-8, 2e-7, \ldots, 2e0, 2e1, \ldots, 2e7, 2e8\}$, $\sigma \in \{2e-8, 2e-7, \ldots, 2e0, 2e1, \ldots, 2e7, 2e8\}$, $\lambda \in \{0.1, 0.2, \ldots, 0.9, 1\}$; |

**Table 4 Two-view learning scenarios.**

| Datasets | Classification tasks | Views (view-A, view-B) | #Sample size |
|---|---|---|---|
| DS1 | AB *vs* CDE | WPD, STFT | 500 |
| DS2 | AB *vs* CDE | WPD, KPCA | 500 |
| DS3 | AB *vs* CDE | STFT, KPCA | 500 |
| DS4 | AB *vs* CD | WPD, STFT | 400 |
| DS5 | AB *vs* CD | WPD, KPCA | 400 |
| DS6 | AB *vs* CD | STFT, KPCA | 400 |
| DS7 | AB *vs* DE | WPD, STFT | 400 |
| DS8 | AB *vs* DE | WPD, KPCA | 400 |
| DS9 | AB *vs* DE | STFT, KPCA | 400 |
| DS10 | AB *vs* CE | WPD, STFT | 400 |
| DS11 | AB *vs* DE | WPD, KPCA | 400 |
| DS12 | AB *vs* CE | STFT, KPCA | 400 |

**Table 5 Classification performance in terms of accuracy on all multi-view learning scenarios.**

| Datasets | KNN_A (KNN on view-A) | KNN_B (KNN on view-B) | SVM_A (SVM on view-A) | SVM_B (SVM on view-B) | SVM-2K | MV-L2-SVM | AMVMED | Proposed model |
|---|---|---|---|---|---|---|---|---|
| DS1 | 0.9098 (0.0019) | 0.9176 (0.0045) | 0.9432 (0.0076) | 0.9521 (0.0087) | 0.9754 (0.0063) | 0.9543 (0.0065) | 0.9643 (0.0043) | **0.9876** (0.0023) |
| DS2 | 0.9213 (0.0032) | 0.9098 (0.0021) | 0.9583 (0.0065) | 0.9321 (0.0087) | 0.9654 (0.0063) | 0.9431 (0.0065) | 0.9546 (0.0043) | **0.9768** (0.0023) |
| DS3 | 0.9223 (0.0034) | 0.9098 (0.0021) | 0.9345 (0.0022) | 0.9321 (0.0087) | 0.9654 (0.0023) | 0.9437 (0.0013) | 0.9554 (0.0063) | **0.9764** (0.0034) |
| DS4 | 0.9214 (0.0034) | 0.9097 (0.0011) | 0.9067 (0.0073) | 0.9164 (0.0027) | 0.9567 (0.0032) | 0.9511 (0.0023) | 0.9598 (0.0044) | **0.9690** (0.0036) |
| DS5 | 0.9214 (0.0034) | 0.9481 (0.0023) | 0.9875 (0.0046) | 0.9467 (0.0056) | **0.9892** (0.0017) | 0.9564 (0.0054) | 0.9578 (0.0023) | 0.9743 (0.0045) |
| DS6 | 0.9324 (0.0052) | 0.9481 (0.0023) | 0.9875 (0.0046) | 0.9467 (0.0056) | 0.9653 (0.0018) | 0.9511 (0.0034) | 0.9587 (0.0033) | **0.9811** (0.0056) |
| DS7 | 0.9331 (0.0026) | 0.9325 (0.0026) | 0.9481 (0.0017) | 0.9435 (0.0037) | 0.9563 (0.0032) | 0.9673 (0.0026) | 0.9543 (0.0046) | **0.9781** (0.0015) |
| DS8 | 0.9331 (0.0026) | 0.9221 (0.0025) | 0.9481 (0.0017) | 0.9387 (0.0026) | 0.9612 (0.0018) | 0.9671 (0.0056) | 0.9409 (0.0055) | **0.9812** (0.0035) |
| DS9 | 0.9631 (0.0015) | 0.9221 (0.0025) | 0.9511 (0.0090) | 0.9387 (0.0026) | 0.9654 (0.0143) | **0.9786** (0.0087) | 0.9765 (0.0049) | 0.9760 (0.0054) |
| DS10 | 0.9318 (0.0079) | 0.9543 (0.0056) | 0.9345 (0.0054) | 0.9245 (0.0064) | 0.9534 (0.0048) | 0.9501 (0.0047) | 0.9534 (0.0019) | **0.9756** (0.0087) |
| DS11 | 0.9134 (0.0078) | 0.9215 (0.0056) | 0.9381 (0.0054) | 0.9275 (0.0034) | 0.9452 (0.0036) | 0.9517 (0.0045) | 0.9732 (0.0017) | **0.9789** (0.0087) |
| DS12 | 0.9532 (0.0035) | 0.9378 (0.0043) | 0.9785 (0.0038) | 0.9634 (0.0014) | 0.9763 (0.0013) | 0.9587 (0.0054) | 0.9661 (0.0064) | **0.9898** (0.0034) |
| Average | 0.9311 | 0.9333 | 0.9472 | 0.9434 | 0.9646 | 0.9561 | 0.9596 | **0.9787** |

**Note:**
Bold entries indicate the best performance achieved by the corresponding method.

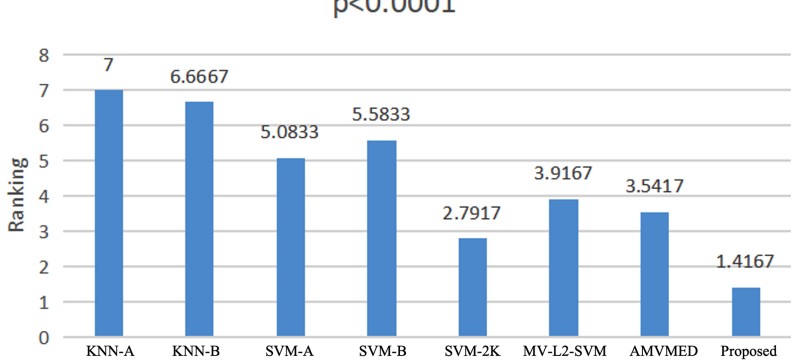

**Figure 5 Friedman rankings of all models.**

To construct a two-view learning scenario, based on "Data", three feature extraction methods, namely wavelet packet decomposition (WPD), short-time Fourier transform (STFT) and kernel principal component analysis (KPCA) are adopted, to extract time-frequency features, frequency-domain features and time-domain features from the original EEG signals, as shown in Fig. 2. Finally, 12 datasets are constructed, as shown in Table 4.

## Experimental results and analysis

The experimental results are reported in Table 5. We can see from Table 5 that the proposed model wins the best performance on most datasets. Only on DS5, DS9, the proposed model performs worse than SVM-2K and MV-L2-SVM. The advantages of the proposed model indicate the promising ability of the shared hidden space. From the promising results, it can be found that by constructing the expanded space and utilizing the information of both the shared hidden space and the original space for learning, thereby fully utilizing the relevant information of samples within and across views, the proposed model effectively solves the problem that the difference between samples of the same class from different views is greater than the difference between samples of different classes from the same view. The experimental results also indicate the power of KDE which is used to construct the shared hidden space.

## Statistical analysis

We use the Friedman test (*Zimmerman & Zumbo, 1993*; *Sakamoto et al., 2015*) to conduct a statistical analysis of the experimental results on all methods across all datasets. The Friedman test is a non-parametric testing method that can be used to analyze whether there are significant differences in performance among multiple methods on multiple datasets. The principle is to first obtain the average ranking of each method's performance on all datasets, and then compare whether these rankings are the same. If they are the same, it indicates that all methods have the same performance, otherwise it suggests that there are significant differences in performance among all methods. If there are significant differences among all methods, we further use a Holm *post-hoc* hypothesis test to specifically analyze which methods and our proposed algorithm have significant differences. From Fig. 5, we see that 2V-SVM-SH wins the best ranking result. The *p*-

**Table 6 Holm test results with α = 0.05.**

| $i$ | Algorithm | $z = (R_0 - R_i)/SE$ | $p$ | $Holm = \alpha/i$ | Hypothesis |
|---|---|---|---|---|---|
| 7 | KNN-A | 5.583333 | 0 | 0.007143 | Rejected |
| 6 | KNN-B | 5.25 | 0 | 0.008333 | Rejected |
| 5 | SVM-B | 4.166667 | 0.000031 | 0.01 | Rejected |
| 4 | SVM-A | 3.666667 | 0.000246 | 0.0125 | Rejected |
| 3 | MV-L2-SVM | 2.5 | 0.012419 | 0.016667 | Rejected |
| 2 | AMVMED | 2.125 | 0.033587 | 0.025 | Not rejected |
| 1 | SVM-2K | 1.375 | 0.169131 | 0.05 | Not rejected |

values embedded in Fig. 5 computed by Friedman test hint that there are significant differences among different models. From Table 6, it can be seen that all hypothesis is rejected except the proposed model *vs* AMVMED and the proposed model *vs* SVM-2K. These results indicate that the proposed model performs significantly better than KNN-A, KNN-B, SVM-B, SVM-A and MV-L2-SVM. Although the hypothesis of the proposed model *vs* AMVMED and the proposed model *vs* SVM-2K is not reject, the low p-value of the proposed model *vs* AMVMED and the proposed model *vs* SVM-2K also indicates the reveal the competition of the proposed model.

## CONCLUSIONS

In this study, a multi-view support vector machine based on a shared hidden space is constructed using kernel density estimation. The method is designed to address the problem of decreased recognition performance due to the difference in sample characteristics between different view models in multi-view learning. The method involves incorporating SVM into the shared hidden space, resulting in an effective solution to the problem of solving the classic QP problem. Experimental results on EEG-based epilepsy diagnosis demonstrate that our proposed method is better able to extract complementary information between different view models than other methods.

In practical applications, annotating training samples is often a time-consuming task. Therefore, in subsequent research, we intend to extend the multi-view algorithm proposed in this article to transfer learning scenarios, aiming to reduce the reliance on labeled samples.

## ACKNOWLEDGEMENTS

We would like to thank all reviewers and the associate editors who give valuable comments.

### Funding

This work was supported by the first batch of industry-university cooperation collaborative education projects in 2021 (No. 202101202002), the Natural Science

Foundation of Colleges and Universities of Anhui Province (No. KJ2020A0773), and the Key Research Project of Natural Science in universities of Anhui Province (No. KJ2019A1307, 2023AH052269). The funders had no role in study design, data collection and analysis, decision to publish, or preparation of the manuscript.

### Grant Disclosures

The following grant information was disclosed by the authors:
First Batch of Industry-university Cooperation Collaborative Education Projects in 2021: 202101202002.
Natural Science Foundation of Colleges and Universities of Anhui Province: KJ2020A0773.
Key Research Project of Natural Science in universities of Anhui Province: KJ2019A1307, 2023AH052269.

### Competing Interests

The authors declare that they have no competing interests.

### Author Contributions

- Xiujian Hu conceived and designed the experiments, performed the experiments, analyzed the data, performed the computation work, prepared figures and/or tables, authored or reviewed drafts of the article, and approved the final draft.
- Yicheng Xie conceived and designed the experiments, performed the experiments, authored or reviewed drafts of the article, and approved the final draft.
- Hui Zhao conceived and designed the experiments, performed the experiments, analyzed the data, authored or reviewed drafts of the article, and approved the final draft.
- Guanglei Sheng conceived and designed the experiments, performed the experiments, authored or reviewed drafts of the article, and approved the final draft.
- Khin Wee Lai analyzed the data, performed the computation work, authored or reviewed drafts of the article, and approved the final draft.
- Yuanpeng Zhang analyzed the data, performed the computation work, authored or reviewed drafts of the article, and approved the final draft.

### Data Availability

The raw data and code are available in the Supplemental Files.
The orginal data is available at The Bonn EEG time series download page:
https://www.upf.edu/web/ntsa/downloads/-/asset_publisher/xvT6E4pczrBw/content/2001-indications-of-nonlinear-deterministic-and-finite-dimensional-structures-in-time-series-of-brain-electrical-activity-dependence-on-recording-regi.

### Supplemental Information

Supplemental information for this article can be found online at http://dx.doi.org/10.7717/peerj-cs.1874#supplemental-information.

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
