# Peer review of "Electroencephalography (EEG) based epilepsy diagnosis via multiple feature space fusion using shared hidden space-driven multi-view learning"

_PeerJ Computer Science, doi:10.7717/peerj-cs.1874_

## Round 0.1 · original submission · Major Revisions

Please address the issues raised by the reviewers and prepare a revised manuscript.

**Language Note:** The review process has identified that the English language must be improved. PeerJ can provide language editing services - please contact us at [email protected] for pricing (be sure to provide your manuscript number and title). Alternatively, you should make your own arrangements to improve the language quality and provide details in your response letter. – PeerJ Staff

Reviewer 1 ·

Basic reporting

no comment

Experimental design

no comment

Validity of the findings

no comment

Additional comments

In this study, the authors proposed a shared hidden space-driven multi-view learning algorithm. The algorithm uses kernel density estimation to construct a shared hidden space and combines the shared hidden space with the original space to obtain an expanded space for multi-view learning.

1. In line 68, please re-organize “Li et al. [1] utilized dual-tree complex discrete wavelet transform and extracted nonlinear features from each component. They used ANOVA to select classification features such as Hurst parameter and fuzzy entropy, and employed SVM for classification.”

2. The third contribution “When optimizing the solution, optimization of the proposed model is transformed to a classic QP problem, which can be solved using existing ready-made optimization methods with good effectiveness and theoretical guarantee.” Is not very significant. Please refine and re-organize.

3. Please indicate the future work of this study in the conclusion section.

4. In line 247, namely “Experimental Results and Analysis”, the authors are suggested to make more analysis regarding the experimental results.

Reviewer 2 ·

Basic reporting

The authors proposed a multiple feature space fusion method using shared hidden space-driven multi-view learning and applied it to EEG-based epilepsy diagnosis. The contribution is significant and the organization is good. But to improve the quality of the manuscript, the authors should consider the following comments.

1. It is better to give time complexity analysis of Algorithm 2.

2. The analysis of the experimental results is too short to highlight the advantages of the proposed methods.

3. In the conclusion section, the authors should give the future work.

4. Please double check all equations in order to avoid typos.

5. “In Section 2, we introduce the EEG data used in this study and the corresponding multiple feature space representation. In Section 3, we present the proposed model. In Section 4, experimental results are reported and in the last section, the whole study is summarized.”
Regarding the above statement, please avoid using Section 2, Section 3, etc. Please replace them by Section of Data, Section of Methodology.

6. The language should be edited by a fluent English speaker.

Experimental design

no comment

Validity of the findings

no comment

Reviewer 3 ·

Basic reporting

To overcome such shortcomings, in this study, a shared hidden feature space method is constructed by using kernel density estimation, and it is extended to an expanded space by combining it with the original space. Then, the support vector machine (SVM) is introduced and a multi-view SVM based on the shared hidden space is proposed to take a careful consideration of the differences and relationships between samples from different views.

1. The abstract should be re-written. The contribution is not very clear from this version.

2. In the first section, more references are expected to highlight the motivations of this study.

3. The conclusion section is too simple. Please enrich the conclusion section including adding the future work.

4. Please analyze the time complexity of the proposed algorithm.

5. Some typos should be corrected.

Experimental design

The experiments are well-designed and the experimental results are convincing.

Validity of the findings

The experimental results are convincing

---

## Round 0.2 · accepted · Accept

I concur with the three reviewers to accept the paper.

Reviewer 1 ·

Basic reporting

no comment

Experimental design

no comment

Validity of the findings

no comment

Additional comments

The authors have addressed my comments well, this version can be accepted.

Reviewer 2 ·

Basic reporting

No comments

Experimental design

No comments

Validity of the findings

No comments

Additional comments

No comments

Reviewer 3 ·

Basic reporting

The article meets the PeerJ criteria and should be accepted as is

Experimental design

no comment

Validity of the findings

no comment

Additional comments

no comment